# Nonlinear Optical Response of Graphene Oxide Langmuir-Blodgett Film as Saturable Absorbers

**DOI:** 10.3390/nano9040640

**Published:** 2019-04-19

**Authors:** Jiang Wang, Yonggang Wang, Taijin Wang, Guangying Li, Rui Lou, Guanghua Cheng, Jing Bai

**Affiliations:** 1School of Physics and Information Technology, Shaanxi Normal University, Xi’an 710119, China; j_wang@snnu.edu.cn (J.W.); taijinwang@snnu.edu.cn (T.W.); 2State Key Laboratory of Transient Optics and Photonics, Xi’an Institute of Optics and Precision Mechanics, Chinese Academy of Sciences, Xi’an 710119, China; li_guang_ying@163.com (G.L.); lourui0423@163.com (R.L.); 3Electronic Information College, Northwestern Polytechnical University, Xi’an 710072, Shaanxi, China; gcheng@opt.ac.cn; 4Department of Physics, Taiyuan Normal University, Taiyuan 030031, China; bai2000yw@163.com

**Keywords:** Langmuir-Blodgett films, nonlinear optical materials, saturable absorbers, graphene oxide

## Abstract

Two-dimensional (2D) materials as saturable absorbers (SAs) have attracted intense interest for applications in pulsed laser generation because of their distinguishing optical properties. However, the existing methods of preparing SAs were insufficient. Here, we fabricated graphene oxide (GO) SAs by Langmuir-Blodgett (LB) methods for passively Q-switched Nd:YAG laser. The GO sheets were deposited on a quartz plate using the LB method. Two different LB-GO SAs grown under the surface pressure of 22 and 38 mN/m were obtained. Compared with the drop coating method, LB-GO SA exhibited more excellent uniformity, larger nonlinear performance and higher optical transparency. By inserting LB-GO SA into the Nd:YAG laser linear cavity, the short pulse duration of 156 ns and the average output power of 1.313 W were obtained. The slope efficiency was as high as 43.7%, due to low loss of the LB-GO SA. Our results illustrated a new way for preparing the SA using the LB technique.

## 1. Introduction

The beginning of a new era of material science has been demonstrated by discovering many new kinds of nanomaterials with superb and novel applications in electronics, sensors, and so on [1,2,3,4]. Nonlinear optics has made outstanding progress in the field of photonics and lasers [5,6,7]. Among nonlinear nanomaterials, graphene oxide, carbon nanotubes, molybdenum disulfide, black phosphorus, zirconium disulfide and many others are widely used as saturable absorbers (SAs) in lasers [8,9,10,11,12,13,14]. Q-switched lasers are very important in applications related to remote sensing, range finding and telecommunications [15,16] because of their compactness, low-cost and flexible designs [17,18]. Performances of SAs such as a high damage threshold, large area, thickness controllability, low insertion losses, and smooth surface are very important for the Q-switched laser performance [19,20]. A rough material surface will lead to optical scattering, which has a significant impact on the properties of the absorbers [21,22,23]. Therefore, nanoscale modification of surface roughness attracted serious research interest with the goal to improve thin films [24].

Methods such as spin coating [25], chemical vapor deposition (CVD) [26], self-assembly, etc. are currently used to obtain graphene or graphene oxide (GO) films [27]. However, the thickness and uniformity of the films fabricated by spin coating generally vary widely. The thickness of CVD-grown film cannot be controlled easily. In addition, catalysts are often used for the CVD growth, which might lead to additional optical non-saturable losses. Thus, only special substrates can be used for the high-temperature crystal growth using CVD. The Langmuir-Blodgett (LB) technique has been widely used for nearly a century to deposit large-area molecular and particular monolayers onto a variety of substrates [28,29]. In addition, the LB technique has been used to obtain solid films of molybdenum disulfide (MoS_2_) [30], and metal nanoparticles [20,31]. However, surface roughness of these films affects their optical properties [32,33]. GO was also used to prepare SAs because of the simple fabrication method and good dispersion [15]. However, rare studies have been done on GO SAs prepared by the LB technique.

We fabricated two kinds of GO films on the quartz substrate under the 22 and 38 mN/m surface pressure using the LB technology and applied them into a passively Q-switched Nd:YAG laser operated at 1064 nm. In comparison to GO SA prepared by the drop coating method, larger area films can be obtained. LB-GO SAs have some other advantages such as good homogeneity, thickness controllability and high optical transmittance. By inserting SA into the Nd:YAG laser cavity, a stable passive Q-switching operation with the slope efficiency as high as 43.7% was successfully obtained. The maximum average output power of this laser was 1.313 W and the corresponding pulse duration was 156 ns.

## 2. Materials and Methods

### 2.1. Materials

For the experiments, deionized (DI) water was used as a sub-phase. Methanol (CH_3_OH, ≤99.5% A.R.) was used for spreading the liquid. Ethanol (C_2_H_5_OH, ≥99.7% A.R.) was used to clean the barrier and the trough, and acetone (CH_3_COCH_3_, ≥99.5% A.R.), sulfuric acid (H_2_SO_4_ A.R.) and hydrogen peroxide (H_2_O_2_ A.R.) were used to clean and prepare the substrates.

### 2.2. Preparation of GO SAs

Liquid phase exfoliation is a simple and effective method for preparing 2D materials [34,35,36,37]. The preparation procedure used in our experiments is schematically shown in Figure 1.

GO pieces shown in Figure 1a were purchased from XF NANO Inc. (NanJing, China). They were weighed and dispersed in DI water to obtain a 2 mg/mL concentration and then ultra-sonicated for 24 h (see Figure 1b). To remove large agglomerates, the GO solution was centrifuged at 5000 rpm for 20 min (see Figure 1c) and the upper half of the GO solution was then dissolved in methanol at 1:4 volume ratio to obtain a stable GO dispersion for LB deposition. After ultra-sonication for 15 min, the resulting dispersion was placed in a glass bottle (see Figure 1d) and was further used to fabricate GO SA on quartz plates (20 mm in diameter and 1 mm thick) using the drop coating and LB methods (Figure 1e). The GO SA was prepared with a computer-controlled device (JML04C1, 2017JM7085, Powereach, Shanghai, China). Film was positioned on both sides of the probe, which caused viscoelastic surface pressure oscillations (see Figure 2a). The trough was cleaned with ethanol and then filled with DI water with pH = 7.0. Representative initial and final surface areas of the films were 145.6 cm^2^ and 14.56 cm^2^, respectively. 1 mL of GO dispersion was spread onto the DI water surface. After the surface was completely covered, the system was left undisturbed for 30 min to allow the material to self-assemble by non-covalent interactions. Then, the barriers were compressed at 7.77 mm/min and the surface pressure changes as function of the area per molecule (π−A) was monitored by a sensor (Wilhelmy plate), as shown in Figure 2b, which illustrates the effect of surface pressure increase on the monomeric area of GO dispersion. π−A curves are typically divided into four distinct regions [38,39]. Region I corresponds to the gaseous phase with molecules being far apart from each other; this region is characterized by weak intermolecular interactions. Region II corresponds to the liquid phase condensate, which is characterized by a small space between insoluble molecules, some intermolecular connectivity characteristics of liquid molecules, and the loose flow structure can still be compressed. Region III corresponds to the condensed solid phases. In this region, the distance between insoluble molecules is small enough, that there is almost no space between the molecules for further compression. These molecules are positioned almost vertically on the surface. Region IV corresponds to the solid collapse phase as the monolayers start collapsing. We selected the following surface pressures: 22 mN/m (corresponds to the sample marked as GO-22, see Figure 1e) and 38 mN/m (corresponds to the sample marked as GO-38). These two pressures correspond to regions II and III, respectively. The films were compressed by the barriers moving at 4.85 mm/min. GO materials were transferred to the quartz plate, which was first processed with a hydrophilic solution (H_2_SO_4_ and H_2_O_2_ mixture with 3:1 volume ratio). The quartz plate was slowly and vertically dipped into the holder containing GO dispersion at 0.5 mm/min rate. After the deposition of the GO LB films, the substrate was placed into a loft drier and was kept there at 80 °C for 10 h. Finally, ultrathin GO-22 and GO-38 LB SA films were obtained.

### 2.3. Characterization of GO SAs

The morphology of the GO SAs was examined using the scanning electron microscopy (Nova NanoSEM Training-X50 series, FEI, Eindhoven, The Netherlands). The GO SAs measurements were performed on a micro-Raman system (obtained using LabRam confocal Microprobe system, Horiba Jobin Yvon, Paris, France) with a 532 nm laser. An atomic force microscope (AFM, Dimension Icon, Bruker Nano Inc., Mannheim, Germany) was employed to observe the microstructures of GO SAs nanostructures. The linear transmission spectra and nonlinear optical absorption of the samples were measured by the spectrophotometer (TU-1810, Persee Universal Instrument Co., Beijing, China) and home-made picosecond pulsed Nd:YAG laser (24 ps and 125 MHz) with the twin-detector measurement technique operated at 1064 nm, respectively.

### 2.4. Laser Cavity

GO-22 SA and GO-38 SA samples were inserted in a laser linear cavity (see Figure 3). The concentration of Nd^3+^ of the Nd:YAG crystal is 1.2 at%. Its dimensions were 3 × 3 × 5 mm^3^ and the crystal was cut in [111] direction. One of the end crystal surfaces was coated with a high transmission film transmitting at 808 nm and a high reflection film reflecting at 1064 nm. One of the end surfaces of the crystal was coated with a high transmission film (transmitting at 808 nm) and a high reflection (HR) film (reflecting at 1064 nm). This crystal was pumped by an 808 nm diode laser (DL) with a 30 W maximum output power. We designed a 1 cm long laser cavity. The output coupler (OC) (R = ∞) was an output coupler with the transmission of 3% at 1064 nm. The laser temperature was maintained at 16 °C by a chilled water-based cooling equipment to keep the output stable. GO SA samples were inserted into the laser cavity close to the OC. A 1 GHz oscilloscope (Rohde & Schwarz, RTO1014, Munich, Germany) is used for the measurement.

## 3. Results and Discussion

### 3.1. Characterization of LB-GO SAs Film

Figure 4 shows SEM images of the sample as well as their Raman spectra. The surface of the GO films fabricated by the drop coating method showed a large accumulation and wrinkles (see Figure 4c). In contrast, GO films prepared by the LB technique, GO-22 (shown in Figure 4a) and GO-38 (shown in Figure 4b) were more uniform. These results agree with the π−A curve, indicating that coatings compressed in the solid region are typically more compact then those compressed in the liquid zone. Raman spectra showed three characteristic peaks: 1D peak at 1356.5 cm^−1^,1G peak at 1604.9 cm^−1^ and 2D peak at 2739.6 cm^−1^ for the GO-22 sample and 1D peak at 1351.1 cm^−1^,1G peak at 1601.7 cm^−1^ and 2D peak at 2738.2 cm^−1^ for the GO-38 sample (see Figure 4d,e, respectively). 1D, 1G and 2D Raman bands associated with the GO film obtained by drop coating were observed at 1346.4 cm^−1^_,_ and 1600.1 cm^−1^ and 2732.6 cm^−1^, respectively (see Figure 4f). Thus, the peak positions for GO coatings obtained using different methods were similar [15,40].

Sizes and thicknesses of the as-prepared GO-22, GO-38 and GO-Drop samples were determined by AFM. 3D topographical images of the GO-22, GO-38 and GO-Drop samples are shown in Figure 5(a1–c1), respectively. Their corresponding 2D images are shown in Figure 5(a2–c2), respectively. To determine film thicknesses, three different sections were chosen for the measurements (see Figure 5(a3,b3,c3)). From Figure 5(a3,b3) we estimated that the thickness of GO in the solvents was 1–6 nm. Since the thickness of a single GO layer is ~1 nm, our prepared films were probably composed of five or more layers of GO. However, the thickness of the GO-Drop film was 20–80 nm (see Figure 5(c3)), and the uniformity of the film was much worse than the film prepared by LB.

### 3.2. Nonlinear Optical Characteristics of LB-GO SA

Linear transmittances of the quartz plate (92.7%), GO-22 SA (85.4%) and GO-38 SA (81.1%) were measured, respectively (see Figure 6a). Their transmittance changes slightly with the increase of the wavelength. However, transmittances of the A and B areas (A~16 mm^2^, B~16 mm^2^) located on the GO-Drop sample surface were 76.2 and 65.5%, respectively, which indicates a significant variation in thickness (as shown in Figure 1e). Experimental data were fitted using the following equation [20]: Τ(I) = 1 − ΔΤ_exp_(−I/I_sat_) − Τ_ns_, where T(I) is the transmission, ΔT is the modulation depth, I is the input intensity, I_sat_ is the saturation intensity and T_ns_ is the non-saturable loss. Values of ΔT, I_sat_, and T_ns_ for the GO-22 SA sample were determined to be 4.86%, 4.26 MW/cm^2^, and 9.06%, respectively. Similar values for the GO-38 SA sample were 10.51%, 2.73 MW/cm^2^ and 8.99%, respectively (see Figure 6b).

### 3.3. LB-GO Q-Switched Laser

We obtained stable passive Q-switching (QS) operation when the pump power exceeded 3.7 W. Q-switching was tested using both GO-22 and GO-38 SAs. Two groups of stable QS operations could be maintained when the pump power varied from 3.7 W to 6.2 W (see Figure 7a). The shortest pulse durations reached 202 and 156 ns when GO-22 and GO-38 SAs were used, respectively (see Figure 7b). The average output power of the continuous wave (CW) laser and QS lasers with SAs of GO-22 and GO-38 were investigated using the same cavity structure (see Figure 8). The average output power of CW and QS increased almost linearly as the pump power increased. The threshold pump powers for CW and QS were 1.55 and 3.7 W, respectively. The slope efficiency of CW laser was 50.7%, and the slope efficiencies of QS with GO-22 and GO-38 SAs were 40.7% and 43.7%, respectively. These values are the highest among other pulsed lasers based on broadband SAs (see Table 1). Absorber films obtained by the LB method were not only thin but also flat, providing the laser with smaller non-saturable losses. Thus, the inserting losses were very small too, which is advantageous for high slope efficiency and large average output power lasers.

The maximum average output power for QS operations with GO-22 and GO-38 SAs were 1.03 W and 1.313 W, respectively, which were obtained under 6.2 W pumping power. The pulse repetition rate increased almost linearly with the incident pump power augmentation, while the single pulse duration decreased (see Figure 8b). When the pump power increased from 5 to 6.2 W, the pulse repetition rate increased from 0.877 to 1.16 MHz and 1.01 to 1.25 MHz, meanwhile, the pulse durations decreased from 350 to 202 ns and from 258 to 156 ns for the GO-22 and GO-38 SAs, respectively. Thus, the performance of the SA fabricated at the 38 mN/m surface pressure was better than the one fabricated at the 22 mN/m surface pressure. The single pulse energy of QS with GO-22 and GO-38 SAs increased almost linearly with the incident pump power (see Figure 8c). The largest single pulse energy of 0.89 and 1.05 μJ for Go-22 and GO-38 SAs were achieved, respectively, when the pump power reached 6.2 W. The largest peak power of Q-switching for GO-22 and GO-38 SAs equal to 4.39 and 6.70 W were achieved when the pump power was 6.2 W (see Figure 8d). Therefore, different modulation depths of absorbers can be obtained by the LB method under different surface pressures, thus achieving a variety of lasers.

## 4. Conclusions

We demonstrated a successful application of LB-fabricated GO SA for a passively Q-switched Nd:YAG laser. GO nanostructures were preassembled and reorganized at the air-water interface using the LB method. Two kinds of SAs were prepared on the different conditions (The surface pressures was 22 and 38 mN/m, respectively). LB-GO SAs exhibited large nonlinear SA properties with the modulation depth of 10.51% at the 1 µm laser wavelength. By inserting the novel GO-38 SA into a Nd:YAG laser linear cavity, Q-switched lasers with the pulse duration as short as 156 ns were obtained with the maximum average output power equal to 1.313 W. The slope efficiency was calculated to be as high as 43.7%, which is the highest in Q-switched lasers with broadband absorbers reported in the literature. Therefore, the LB method may be a good choice to make practical 2D materials absorbers.

## Figures and Tables

**Figure 1 nanomaterials-09-00640-f001:**
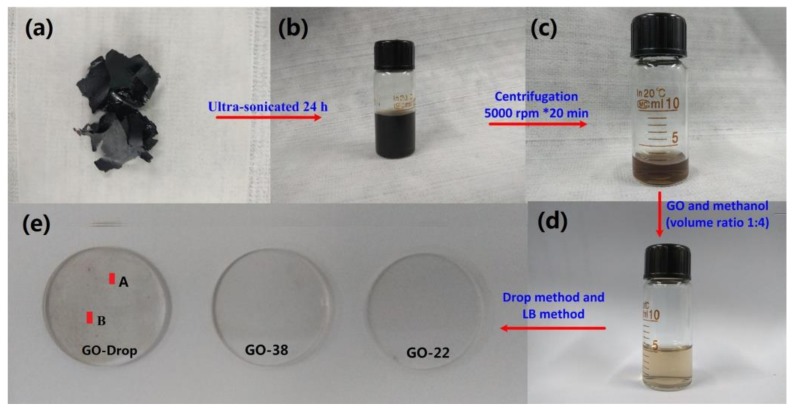
Preparation steps of graphene oxide (GO) solution: (**a**) GO slices, ultrasonicated (**b**) and centrifuged (**c**) GO solution, (**d**) GO dispersion of Langmuir-Blodgett (LB) in methanol, and (**e**) the GO saturable absorber (SA) films coated on the quartz plate. The following surface pressures were selected: 22 mN/m (corresponds to the sample marked as GO-22) and 38 mN/m (corresponds to the sample marked as GO-38), these two pressures correspond to regions II and III, respectively.

**Figure 2 nanomaterials-09-00640-f002:**
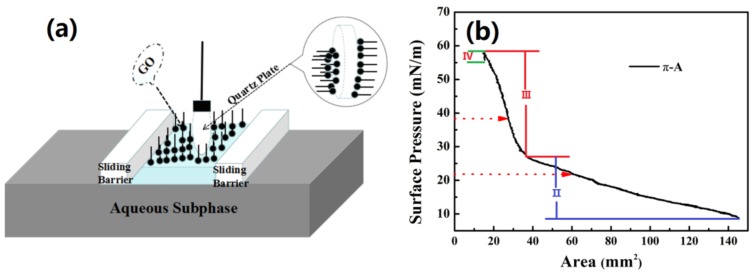
(**a**) Schematic of the Langmuir-Blodgett experimental setup; (**b**) surface pressure versus area (π−A) isotherm of GO.

**Figure 3 nanomaterials-09-00640-f003:**
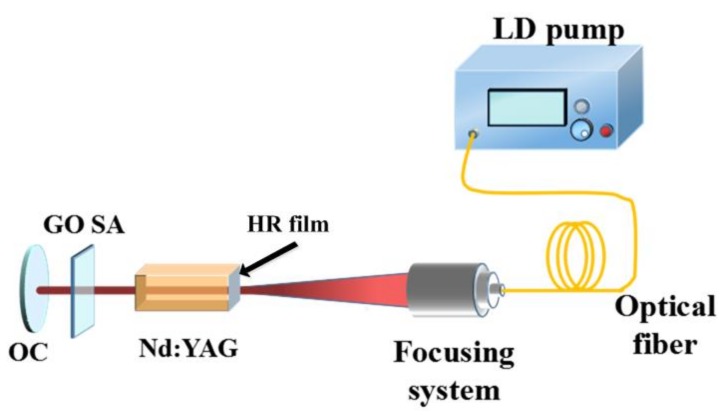
Experimental setup of passively Q-switched (QS) laser with GO SAs.

**Figure 4 nanomaterials-09-00640-f004:**
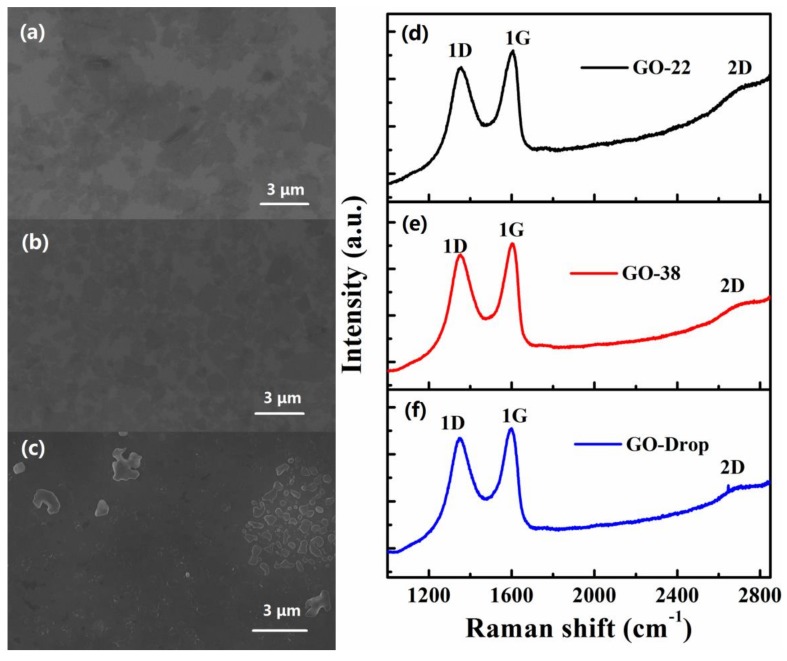
SEM image of GO SA samples prepared by the LB method (samples GO-22 (**a**) and GO-38 (**b**)) and by the drip method (sample GO-Drop (**c**)). Raman spectra of the GO-22 (**d**), GO-38 (**e**) and GO-Drop (**f**) samples.

**Figure 5 nanomaterials-09-00640-f005:**
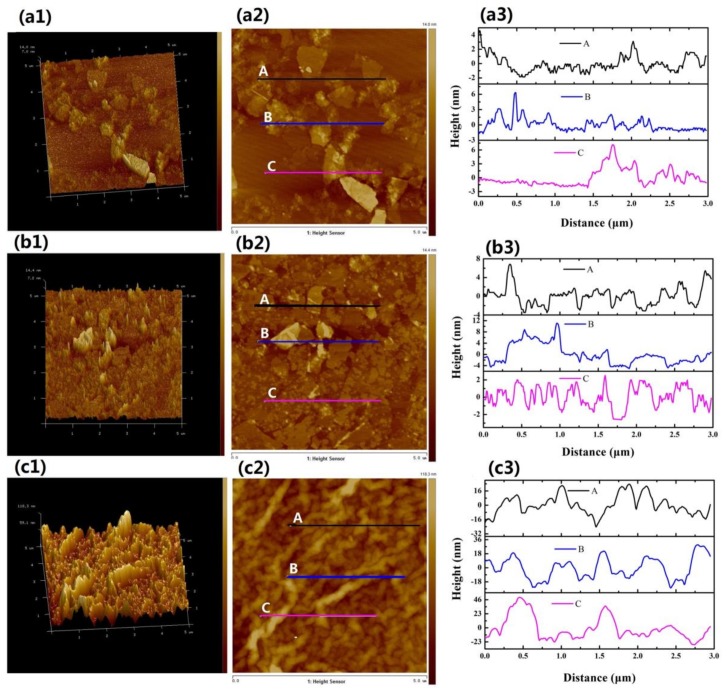
Atomic force microscope (AFM) of GO SA films GO-22 (**a**), GO-38 (**b**) and GO-Drop (**c**). (**a1**,**b1**,**c1**) are 3D topographical images; their corresponding 2D topographical images are shown in (**a2**,**b2**,**c2**), respectively; thickness profiles are shown in (**a3**,**b3**,**c3**), respectively.

**Figure 6 nanomaterials-09-00640-f006:**
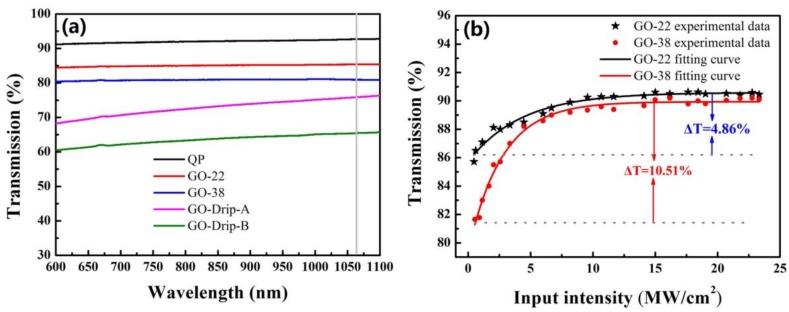
(**a**) Linear transmission spectra and (**b**) nonlinear optical absorption of GO-22 SA and GO-38 SA, QP refers to quartz plate.

**Figure 7 nanomaterials-09-00640-f007:**
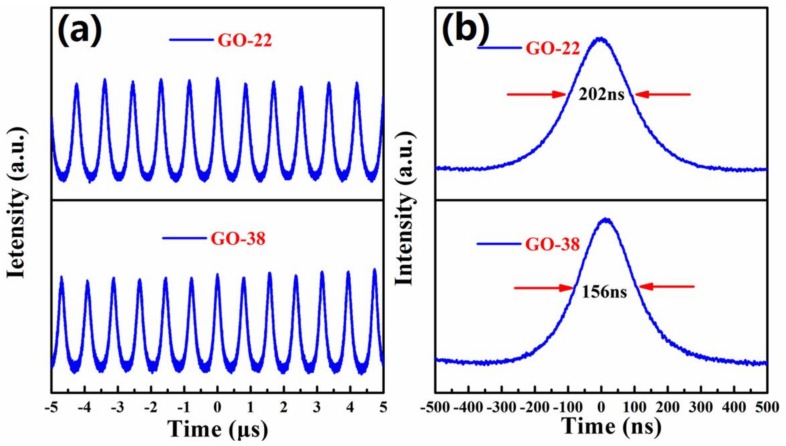
(**a**) Temporal pulse trains of Q-switched (QS) laser and (**b**) the single pulse profiles at 6.2 W pump power obtained with GO-22 and GO-38 SAs, respectively.

**Figure 8 nanomaterials-09-00640-f008:**
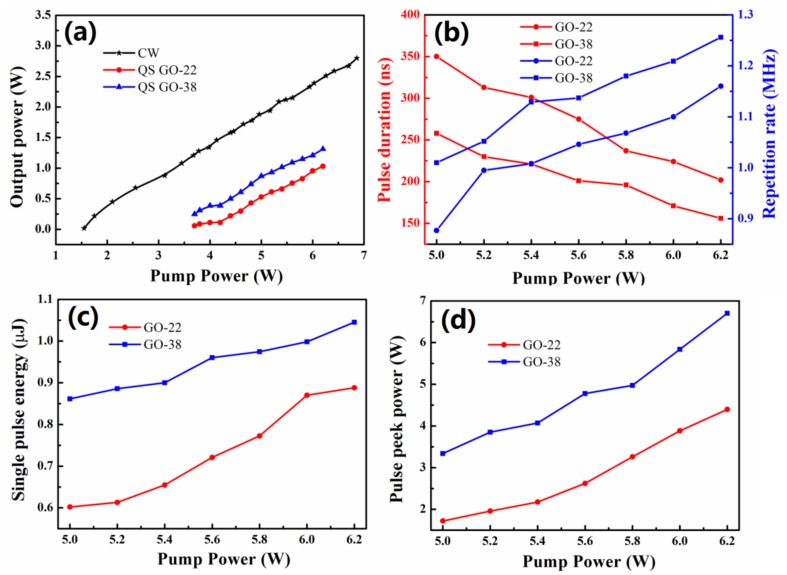
(**a**) Average output power of CW and QS as a function of the pump power; (**b**) repetition rates and pulse durations as a function of the pump power. Single pulse energy (**c**) and peak power (**d**) from passively Q-switched Nd:YAG laser versus incident pump power with GO-22 and GO-38 SAs.

**Table 1 nanomaterials-09-00640-t001:** Q-switched solid-state pulsed lasers based on SAs.

SA Type	Laser Type	Η (%)	Λ (nm)	Τ (ns)	P (W)	E (µJ)	Frep (kHz)	Ref.
Bi_2_Te_3_	Yb:GAB	24.7	1064	303	0.213	1.2	178.2	[41]
Bi_2_Te_3_	Yb:KGW	8.8	1041	1600	0.439	2.64	166.7	[42]
Graphene	Nd:GdVO_4_	37	1063	105	2.3	3.2	704	[43]
Graphene	Nd:YAG	-	1064	161	0.105	0.159	660	[44]
MoS_2_	Yb:LGGG	24	1025.2	182	0.6	1.8	333	[45]
MoS_2_	Nd:YAlO_3_	38.4	1079.5	227	0.26	1.11	232.5	[46]
GO	Nd:GdVO_4_	17	1064	104	1.22	2	600	[15]
Graphene	Nd:YAG	7.8	1123	875.7	0.332	-	46.8	[47]
GO-22	Nd:YAG	40.7	1064	202	1.03	0.89	1160	Our work
GO-38	Nd:YAG	43.7	1064	156	1.313	1.04	1256	Our work

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
