# Peer review of "Nonlinear Optical Response of Graphene Oxide Langmuir-Blodgett Film as Saturable Absorbers"

_nanomaterials, 2019, doi:10.3390/nano9040640_

Reviewer 1 Report

The work of Wang et al. is devoted to the actual topic of new materials developing for saturable absorbers creating. It is well known that graphene oxide could be used as a saturable absorber; moreover, the Langmuir-Blodgett method is widely used to graphene oxide films formation. Nevertheless, it is not known previously works focused on both - graphene oxide films preparation by the Langmuir-Blodgett method and creating a saturable absorber from those films. The article may be accepted for publication after minor revision.

1.      On page 3 (lines 102 - 111), the authors use the terms Region 1-4 in describing the pressure versus area curve. However, to understand the meaning of these Regions, the reader needs to use the text of the original articles at the indicated links, which complicates article perception. Authors need to mark the boundaries of Regions 1-4 in Figure 2b.

2.      What is the spectral resolution of the Raman spectrometer and what is the error in determining the position of the bands maxima? Judging from the spectra presented in Figure 4b, the values of the bands maxima are presented on page 5 (lines 155-157) with an excessive accuracy of 1 cm-1.

3.      The authors discuss the transmission at points A and B of the GO-Drop sample on page 7 (lines 187-189). However, the spectra cannot be measured from the point, it does not have a physical meaning, but is carried out from an area of finite size. It is necessary to specify the size of the measured areas A and B.

4.      In conclusion (line 241), the authors claim that two saturable absorbers with “different states” have been prepared. It is not clear what is "different states". Please, give an explanation.

Author Response

Dear Editor and Reviewers,

Thanks for your letter of “Decision for Nanomaterials manuscript ID: nanomaterials-483820” and the reviewers’ thoughtful suggestions concerning our manuscript entitled “Nonlinear Optical Response of Graphene oxide Langmuir-Blodgett Film as Saturable Absorbers”. Those comments and suggestions are very valuable and helpful for revising and improving our paper, as well as the important guiding significance to our researches. We have made careful modifications to this manuscript. All revised portions are highlighted in red in this paper. We hope that it has reached your magazine’s standard for publication. The main corrections in paper and the responds to the reviewer’s comments are as follows.

Best Regards

Yonggang Wang & Jiang Wang

Response to Reviewer 

For Comments of Reviewer:

The work of Wang et al. is devoted to the actual topic of new materials developing for saturable absorbers creating. It is well known that graphene oxide could be used as a saturable absorber; moreover, the Langmuir-Blodgett method is widely used to graphene oxide films formation. Nevertheless, it is not known previously works focused on both - graphene oxide films preparation by the Langmuir-Blodgett method and creating a saturable absorber from those films. The article may be accepted for publication after minor revision.

Our reply to Reviewer:

Question 1: On page 3 (lines 102-111), the authors use the terms Region 1-4 in describing the pressure versus area curve. However, to understand the meaning of these Regions, the reader needs to use the text of the original articles at the indicated links, which complicates article perception. Authors need to mark the boundaries of Regions 1-4 in Figure 2b.

Answer to question 1: Thank you for your suggestion. The π-A curve shows three regions (Ⅱ, Ⅲ and Ⅳ) in our experiment due to the influence of different solution concentration, solution quantity and other factors. In addition, Ⅱ and Ⅲ are our study emphasis. In order to better explain the significance of different areas of the LB curve, we mark the boundaries of Regions Ⅱ-Ⅳ.

It is modified as follows:

Figure 2. (a) Schematic of the Langmuir-Blodgett experimental setup, (b) surface pressure vs area (π−A) isotherm of GO

Question 2: What is the spectral resolution of the Raman spectrometer and what is the error in determining the position of the bands maxima? Judging from the spectra presented in Figure 4b, the values of the bands maxima are presented on page 5 (lines 155-157) with an excessive accuracy of 1 cm-1.

Answer to question 2: Thank you for your advice. The spectral resolution of the Raman spectrometer ≦1 cm-1. There are many factors that affect the results of Raman test, such as test environment, testers, Raman optical path, and so on. We measured the samples in the previous experiment again and got good results. We have revised it in the manuscript.

It is modified as follows:

Figure 4. SEM image of GO SA samples prepared by LB method (samples GO-22 (a) and GO-38 (b)) and by the drip method (sample GO-Drop (c)). Raman spectra of the GO-22 (d), GO-38 (e) and GO-Drop (f) samples

Question 3: The authors discuss the transmission at points A and B of the GO-Drop sample on page 7 (lines 187-189). However, the spectra cannot be measured from the point, it does not have a physical meaning, but is carried out from an area of finite size. It is necessary to specify the size of the measured areas A and B.

Answer to question 3: Thank you for your suggestion. According to the light spot size of spectrophotometer (TU-1810, China) in the sample is about 16 mm2, so the area of both A and B is about 16 mm2.

It is modified as follows:

Linear transmittances of quartz plate (92.7 %), GO-22 SA (85.4 %) and GO-38 SA (81.1%) were measured, respectively (see Figure 6a). Their transmittance changes slightly with the increase of the wavelength. However, transmittances of the A and B areas (A~16 mm2, B~16 mm2) located on the GO-Drop sample surface were 76.2 and 65.5%, respectively, which indicates significant variation in thickness (as shown in Figure 1e).

Figure 1. Preparation steps of GO solution: (a) GO slices, ultrasonicated (b) and centrifuged (c) GO solution, (d) GO dispersion of LB in methanol, and (e) the GO SA films coated on the quartz plate. The following surface pressures was selected: 22 mN/m (corresponds to the sample marked as GO-22) and 38 mN/m (corresponds to the sample marked as GO-38),these two pressures correspond to regions II and III, respectively.

Question 4: In conclusion (line 241), the authors claim that two saturable absorbers with “different states” have been prepared. It is not clear what is "different states". Please, give an explanation.

Answer to question 4: Great thanks for the suggestion of reviewer. The surface pressure at 22 mN/m and 38 mN/m corresponds to the membrane in two states. We selected the following surface pressures: 22 mN/m (corresponds to the sample marked as GO-22) and 38 mN/m (corresponds to the sample marked as GO-38). These two pressures correspond to regions II and III, respectively.

It is modified as follows:

Two kinds of SAs were prepared on the different conditions (The surface pressures was 22 and 38 mN/m respectively).

We appreciate for Editors/Reviewers’ warm work earnestly, and hope that the correction will meet with approval. Once again, thank you very much for your comments and suggestions.

Reviewer 2 Report

The manuscript entitled as "Nonlinear optical response of graphene oxide Langmuir-Blodgett film as saturable absorbers" experimentally demonstrated graphene oxide saturable absorbers (based on LB technique) for the use of passively Q-switched Nd:YAG laser. The proposed modality exhibited large nonlinear absorptance and transmittance. This manuscript has a strong potential for a second review after applying the issues and addressing the shortcomings listed below (even if this work similar to the authors’ recent work: Chin. Opt. Lett. 17, 020009 (2019)):

1.      The authors should polish/revise some grammatical mistakes and typos along the manuscript. For instance, ‘…intensity and non-saturable loss…’, ‘…these novel LB-GO SA into…’, ‘…slope efficiency was…’, ‘…our results illustrated potential…’, ‘…preparing the SA using…’, ‘…electrocatalysis, optoelectronics…’, ‘…finding, telecommunications…’, ‘…low cost andflexible designs’, ‘…depth and damage threshold’, ‘Exploring new technique…’, ‘…with smooth surface…’, ‘…thickness vary significantly…’, ‘…have flat surface. But, the film…’, ‘Go is a derivative…’, ‘…which is conducive to the…’, ‘LB-GO SA have other…’, ‘controllability and high optical…’, ‘To the best of our knowledge… (too much repetition along the manuscript). The authors should use a different statement’, ‘…dichalcogenide and black phosphorous’, ‘do not need to use whole version of DI in line 82’, ‘Dimension of the trough…’, ‘…earlier reports.34,35’, ‘…smaller, and, thus, the…’, ‘do not need to use whole version of SEM in line 150’, ‘…theas-prepared…’, ‘…samples was determined…’, ‘…b1 and c1…’, ‘…GO-38 and GO-Drop…’, ‘…of the samples determined…’, ‘…and 4c3…’, ‘…4 nm and between 2 and 6 nm…’, ‘…actual samples thickness…’, ‘…was muchworse…’, ‘… are shown… (too much repetition along the manuscript). The authors should utilize synonyms’, ‘Transmittances of quartz plate…’, ‘…GO-22 SA and GO-38 SA…’.

2.      The following statements should be revised grammatically: ‘The results show that large area GO…’, ‘Performance of SAs such as…’, ‘the second sentence in Section 2.1’.

3.      Proper references should be increased for the following statements: ‘…remote sensing, range finding…’, ‘…LB technique has been used to obtain…’, ‘for the sentence starting from line 57’.

4.      The related references of each equation should be mentioned along the manuscript, if the formulas are taken from some other works.

5.      In the Introduction section (at some points), the transition between sentences are not cohesive. The authors should work on this section and correct them carefully (i.e. between line 52 and 53).

6.      The Introduction part is weak in terms the recent nonlinear optical advancements in the field of photonics and lasers. The authors should shortly discuss them either in the Introduction section or in Section 3.2. The following works should be properly mentioned and cited within the manuscript (the authors can also add another references in addition to the given ones): [(i) Nature 521, 498-502 (2015); (ii) Nano Lett. 19, 605-611 (2019); (iii) Phys. Rev. A 92, 053820 (2015); (iv) Opt. Mater. 73, 729-735 (2017)].

7.      In Figure 1e, what are ‘A’ and ‘B’? They should be mentioned in the caption (even the authors mentioned in line 189). Besides, the meaning of ‘GO-38’ and ‘GO-22’ should be given in the caption.

8.      What is the ‘JML04C1’ system that is used for LB? For a broad range of readers, it should be cleared with giving proper references.

9.      In Figure 2b, the related regions should be indicated, and Figure 3 should be updated based on the information giving in Section 2.4. 

10.  In Figure 4c-f, the corresponding x-axis should be corrected. In addition to the given D and G peaks, the authors should also explore the 2D peak of the prepared samples in the Raman spectra.   

11.  There is no need to use the following statement again and again in the manuscript: ‘The laser source was a home-made mode-locked…’.

12.  Why the repetition rate values in Figure 8b seem lower?

Author Response

Dear Editor and Reviewers,

Thanks for your letter of “Decision for Nanomaterials manuscript ID: nanomaterials-483820” and the reviewers’ thoughtful suggestions concerning our manuscript entitled “Nonlinear Optical Response of Graphene oxide Langmuir-Blodgett Film as Saturable Absorbers”. Those comments and suggestions are very valuable and helpful for revising and improving our paper, as well as the important guiding significance to our researches. We have made careful modifications to this manuscript. All revised portions are highlighted in red in this paper. We hope that it has reached your magazine’s standard for publication. The main corrections in paper and the responds to the reviewer’s comments are as follows.

Best Regards

Yonggang Wang & Jiang Wang

Response to Reviewer 

For Comments of Reviewer:

The manuscript entitled as "Nonlinear optical response of graphene oxide Langmuir-Blodgett film as saturable absorbers" experimentally demonstrated graphene oxide saturable absorbers (based on LB technique) for the use of passively Q-switched Nd:YAG laser. The proposed modality exhibited large nonlinear absorptance and transmittance. This manuscript has a strong potential for a second review after applying the issues and addressing the shortcomings listed below (even if this work similar to the authors’ recent work: Chin. Opt. Lett. 17, 020009 (2019)):

Our Reply:

Thank you for your comments and corrections. In this paper, we mainly studied the preparation process of GO thin films by LB technique. The properties of thin films prepared in two states are compared in detail. Compared with the authors’ recent work: Chin. Opt. Lett. 17, 020009 (2019), there are great differences, such as laser cavity, absorber base and so on.

Question 1: The authors should polish/revise some grammatical mistakes and typos along the manuscript. For instance, ‘…intensity and non-saturable loss…’, ‘…these novel LB-GO SA into…’, ‘…slope efficiency was…’, ‘…our results illustrated potential…’, ‘…preparing the SA using…’, ‘…electrocatalysis, optoelectronics…’, ‘…finding, telecommunications…’, ‘…low cost andflexible designs’, ‘…depth and damage threshold’, ‘Exploring new technique…’, ‘…with smooth surface…’, ‘…thickness vary significantly…’, ‘…have flat surface. But, the film…’, ‘Go is a derivative…’, ‘…which is conducive to the…’, ‘LB-GO SA have other…’, ‘controllability and high optical…’, ‘To the best of our knowledge… (too much repetition along the manuscript). The authors should use a different statement’, ‘…dichalcogenide and black phosphorous’, ‘do not need to use whole version of DI in line 82’, ‘Dimension of the trough…’, ‘…earlier reports.34,35’, ‘…smaller, and, thus, the…’, ‘do not need to use whole version of SEM in line 150’, ‘…theas-prepared…’, ‘…samples was determined…’, ‘…b1 and c1…’, ‘…GO-38 and GO-Drop…’, ‘…of the samples determined…’, ‘…and 4c3…’, ‘…4 nm and between 2 and 6 nm…’, ‘…actual samples thickness…’, ‘…was muchworse…’, ‘… are shown… (too much repetition along the manuscript). The authors should utilize synonyms’, ‘Transmittances of quartz plate…’, ‘…GO-22 SA and GO-38 SA…’.

Answer to question 1: Great thanks for the suggestion of reviewer. We have revised it highlighted in red in the manuscript. 

It is modified as follows:

Abstract: Two-dimensional (2D) materials as saturable absorbers (SAs) have attracted intense interest for applications in pulsed laser generation because of their distinguishing optical properties. However, the existing methods of preparing SAs were insufficient. Here, we fabricated graphene oxide (GO) SAs by Langmuir-Blodgett (LB) methods for passively Q-switched Nd:YAG laser. The GO sheets were deposited on quartz plate using LB method. Two different LB-GO SAs grown under the surface pressure of 22 and 38 mN/m were obtained. Compared with the drop coating method, LB-GO SA exhibited more excellent uniformity, larger nonlinear performance and higher optical transparency. By inserting LB-GO SA into the Nd:YAG laser linear cavity, the short pulse duration of 156 ns and the average output power of 1.313 W were obtained. The slope efficiency was as high as 43.7%, due to low loss of the LB-GO SA. Our results illustrated a new way for preparing the SA using LB technique. 

1. Introduction

The beginning of a new era of material science has been demonstrated by discovering many new kinds of nanomaterials with superb and novel applications in electronics, sensors, and so on [1-4]. Nonlinear optics has made outstanding progress in the field of photonics and lasers [5-7]. Among nonlinear nanomaterials, graphene oxide, carbon nanotubes, molybdenum disulfide, black phosphorus, zirconium disulfide and many others are widely used as saturable absorbers (SAs) in lasers [8-14]. Q-switched lasers are very important in applications related to remote sensing, range finding, telecommunications [15,16] because of their compactness, low-cost and flexible designs [17, 18]. Performance of SAs such as high damage threshold, large area, thickness controllability, low insertion losses, and smooth surface are very important for the Q-switched laser performance [19, 20]. Rough material surface will lead to optical scattering, which has a significant impact on the properties of the absorbers [21-23]. Therefore, nanoscale modification of surface roughness attracted significant research interest with the goal to improve thin films [24].

The GO SA were prepared with a computer-controlled device (JML04C1, 2017JM7085, Powereach, China). Film was positioned on both sides of the probe, which caused viscoelastic surface pressure oscillations (see Figure 2(a)). The trough was cleaned with ethanol and then filled with DI water with pH = 7.0. Representative initial and final surface areas of the films were 145.6 cm2 and 14.56 cm2, respectively. 1 ml of GO dispersion was spread onto the DI water surface.

Question 2The following statements should be revised grammatically: ‘The results show that large area GO…’, ‘Performance of SAs such as…’, ‘the second sentence in Section 2.1’.

Answer to question 2: Thanks for the reviewer’s suggestion. We have revised it in the manuscript.

 It is modified as follows:

The GO sheets were deposited on quartz plate using LB method. Two different states of the LB-GO SA under the 22 and 38 mN/m surface pressure were obtained. Compared with the drop coating method, LB-GO SA exhibited excellent uniformity, large nonlinear performance and high optical transparency. By inserting LB-GO SA into the Nd:YAG laser linear cavity, the short pulse duration of 156 ns and the average output power of 1.313 W were obtained. The slope efficiency was as high as 43.7%, due to low loss of the LB-GO SA. Our results illustrated that an new way for preparing the SA using LB technique.

Performances of SAs such as high damage threshold, large area, thickness controllability, low insertion losses, and smooth surface are very important for the Q-switched laser performance [19, 20]. Rough material surface will lead to optical scattering, which has a significant impact on the properties of the absorbers [21-23]. Therefore, nanoscale modification of surface roughness attracted significant research interest with the goal to improve thin films [24]. 

3.2. Nonlinear optical characteristics of LB-GO SA

Linear transmittances of quartz plate (92.7 %), GO-22 SA (85.4 %) and GO-38 SA (81.1%) were measured, respectively (see Figure 6a). Their transmittance changes slightly with the increase of the wavelength.

Question 3:  Proper references should be increased for the following statements: ‘…remote sensing, range finding…’, ‘…LB technique has been used to obtain…’, ‘for the sentence starting from line 57’.

Answer to question 3: Thank you for your comments. We have revised it in the manuscript.

It is modified as follows:

  Q-switched lasers are very important in applications related to remote sensing, range finding, telecommunications [15,16]

In recent years, LB technique has been used to obtain solid films of molybdenum disulfide (MoS2) [30], and metal nanoparticles [20,31]

16 Sirota, M.; Galun, E.; Sashchiuk, A.; Krupkin, V.; Glushko, A.; Lifshitz, E. IV-VI semiconductor nanocrystals for passive Q-switching of eye-safe laser, Proceedings of SPIE-The International Society for Optical Engineering 2003, 4970, 53-60.

20. Wang, X.; Wang, Y. G.; Gu, Y. Z.; Li, L.; Wang, J.; Yang, X. G.; Chen, Z. D. Titanium Dioxide Langmuir–Blodgett Film Saturable Absorber for Passively Q-switched Nd:GdVO4 Laser, IEEE Photonics J. 2019, 11,1501110.

Question 4The related references of each equation should be mentioned along the manuscript, if the formulas are taken from some other works.

Answer to question 4: Thanks for the reviewer’s suggestion. We have revised it in the manuscript.

It is modified as follows:

Experimental data were fitted using the following equation [20]: Τ(I)=1-ΔΤexp(-I/Isat)-Τns, where T(I) is the transmission, ΔT is the modulation depth, I is the input intensity, Isat is the saturation intensity and Tns is the non-saturable loss.

20. Wang, X.; Wang, Y. G.; Gu, Y. Z.; Li, L.; Wang, J.; Yang, X. G.; Chen, Z. D. Titanium Dioxide Langmuir–Blodgett Film Saturable Absorber for Passively Q-switched Nd:GdVO4 Laser, IEEE Photonics J. 2019, 11,1501110.

Question 5: In the Introduction section (at some points), the transition between sentences are not cohesive. The authors should work on this section and correct them carefully (i.e. between line 52 and 53).

Answer to question 5: Thanks for the reviewer’s comments.. We have revised it in the manuscript.

It is modified as follows:

Langmuir-Blodgett (LB) technique has been widely used for nearly a century to deposit large-area molecular and particulate mono layers onto a variety of substrates [28, 29]. In addition, LB technique has been used to obtain solid films of molybdenum disulfide (MoS2) [30], and metal nanoparticles [20,31].

Question 6: The Introduction part is weak in terms the recent nonlinear optical advancements in the field of photonics and lasers. The authors should shortly discuss them either in the Introduction section or in Section 3.2. The following works should be properly mentioned and cited within the manuscript (the authors can also add another references in addition to the given ones): [(i) Nature 521, 498-502 (2015); (ii) Nano Lett. 19, 605-611 (2019); (iii) Phys. Rev. A 92, 053820 (2015); (iv) Opt. Mater. 73, 729-735 (2017)].

Answer to question 6: Thanks for the reviewer’s suggestion. We have revised it in the manuscript.

It is modified as follows:

The beginning of a new era of material science has been demonstrated by discovering many new kinds of nanomaterials with superb and novel applications in electronics, sensors, and so on [1-4]. Nonlinear optics has made outstanding progress in the field of photonics and lasers [5-7].

5. Christensen, B. T. R.; Henriksen, M. R.; Schaffer, S. A.; Westergaard, P. G.; Tieri, D.; Ye, J.; Holland, M. J.;  Thomsen, J. W. Nonlinear spectroscopy of Sr atoms in an optical cavity for laser stabilization, Phys. Rev. A 2015, 92, 053820.

6. Luu, T. T.; Garg, M.; Kruchinin, S. Y.; Moulet, A.; Hassan, M. Th.; Goulielmakis, E. Extreme ultraviolet high-harmonic spectroscopy of solids, Nature 2015, 521, 498-502.

7. Ahmadivand, A.; Semmlinger, M.; Dong, L. L.; Gerislioglu, B.; Nordlander, P.; Halas, N. J. Toroidal Dipole-Enhanced Third Harmonic Generation of Deep Ultraviolet Light Using Plasmonic Meta-atoms, Nano Lett. 2019,19, 605-611.

Question 7In Figure 1e, what are ‘A’ and ‘B’? They should be mentioned in the caption (even the authors mentioned in line 189). Besides, the meaning of ‘GO-38’ and ‘GO-22’ should be given in the caption.

Answer to question 7: Thanks for the reviewer’s suggestion. We have revised it in the manuscript.

It is modified as follows:

However, transmittances of the A and B areas (A~16 mm2, B~16 mm2) located on the GO-Drop sample surface were 76.2 and 65.5%, respectively, which indicates significant variation in thickness (as shown in Figure 1e).

We selected the following surface pressures: 22 mN/m (corresponds to the sample marked as GO-22, see Figure 1e) and 38 mN/m (corresponds to the sample marked as GO-38).

Figure 1. Preparation steps of GO solution: (a) GO slices, ultrasonicated (b) and centrifuged (c) GO solution, (d) GO dispersion of LB in methanol, and (e) the GO SA films coated on the quartz plate. The following surface pressures was selected: 22 mN/m (corresponds to the sample marked as GO-22) and 38 mN/m (corresponds to the sample marked as GO-38),these two pressures correspond to regions II and III, respectively.

Question 8: What is the ‘JML04C1’ system that is used for LB? For a broad range of readers, it should be cleared with giving proper references.

Answer to question 8: Great thanks for the suggestion of reviewer. We have revised it in the manuscript.

It is modified as follows:

The GO SA were prepared with a computer-controlled device (JML04C1, 2017JM7085, Powereach, China).

Question 9:  In Figure 2b, the related regions should be indicated, and Figure 3 should be updated based on the information giving in Section 2.4. 

Answer to question 9:Thanks for the reviewer’s suggestion. We have revised it in the manuscript.

It is modified as follows:

Figure 2. (a) Schematic of the Langmuir-Blodgett experimental setup, (b)surface pressure vs area (π−A) isotherm of GO

Figure 3. Experimental setup of passively Q-switched laser with GO SAs

Question 10In Figure 4c-f, the corresponding x-axis should be corrected. In addition to the given D and G peaks, the authors should also explore the 2D peak of the prepared samples in the Raman spectra.  

Answer to question 10: Great thanks for the suggestion of reviewer. In Figure 4c-f, the corresponding x-axis has been corrected. We expand the data scope and find that its 2D peak is not obvious, but it is consistent with the authors’ recent work:Nanotechnology 2011, 22, 455203. We have changed it in the manuscript.

It is modified as follows:

Figure 4. SEM image of GO SA samples prepared by LB method (samples GO-22 (a) and GO-38 (b)) and by the drip method (sample GO-Drop (c)). Raman spectra of the GO-22 (d), GO-38 (e) and GO-Drop (f) samples

Question 11: There is no need to use the following statement again and again in the manuscript: ‘The laser source was a home-made mode-locked…’.

Answer to question 11: Thanks for the reviewer’s suggestion. We have revised the original manuscript, and ‘The laser source was a home-made mode-locked…’. only appears in part 2.3.

It is modified as follows:

2.3 Characterization of GO SAs 

The morphology of the GO SAs was examined using scanning electron microscopy (Nova Nano SEM Training-X50 series). The GO SAs measurements were performed on amicro-Raman system (obtained using LabRam confocal Microprobe system)with a 532 nm laser. An atomic force microscop (AFM, Dimension Icon, Bruker Nano Inc.) was employed to observe the microstructures of GO SAs nanostructures. Linear transmission spectra and nonlinear optical absorption of the samples were measured by the spectrophotometer (TU-1810, China) and home-made picosecond pulsed Nd:YAG laser (24 ps and 125 MHz) with twin-detector measurement technique operated at 1064 nm, respectively.

Question 12: Why the repetition rate values in Figure 8b seem lower?

Answer to question 12:

With the increase of pump power, pulse repetition rate increased from 0.877 to 1.16 MHz and 1.01 to 1.25 MHz, as shown in figure 2. As can be seen from table 1, The repetition rate value is not very low.

Table 1. Q-switched solid-state pulsed lasersbased on SAs

SA type

Laser type

η(%)

λ(nm)

τ(ns)

P(W)

E(µJ)

Frep(kHz)

Ref.

Bi2Te3

Bi2Te3

Graphene

Graphene

MoS2

MoS2

GO

GO

GO-22

GO-38

Yb:GAB

Yb:KGW

Nd:GdVO4

 Nd:YAG

Yb:LGGG

Nd:YAlO3

Nd:GdVO4

Nd:YAG

Nd:YAG

Nd:YAG

24.7

8.8

37

--

24

38.4

17

7.8

40.7

43.7

1064

1041

1063

1064

1025.2

1079.5

1064

1123

1064

1064

303

1600

105

161

182

227

104

875.7

202

156

0.213

0.439

2.3

0.105

0.6

0.26

1.22

0.332

1.03

1.313

1.2

2.64

3.2

0.159

1.8

1.11

2

--

0.89

1.04

178.2

166.7

704

660

333

232.5

600

46.8

1160

1256

41

42

43

44

45

46

15

43

Our work

Our work

  We appreciate for Editors/Reviewers’ warm work earnestly, and hope that the correction will meet with approval. Once again, thank you very much for your comments and suggestions.

Round  2

Reviewer 1 Report

The article is recommended for publication in present form.

Reviewer 2 Report

In its current form, the revised manuscript can be acceptable for MDPI Nanomaterials.